# Study on Design of Non-Circular Gears for Speed Control of the Squid Belly Opening and Gutting Machine (SBOGM)

**Hyo-Seong Jang, Chang-Hyun Lee, Gun-Young Park and Chul Kim ***

School of Mechanical Engineering, Pusan National University, Pusan 46241, Korea;
hyseo5807@gmail.com (H.-S.J.); rksckdgus@naver.com (C.-H.L.); pgy323@gmail.com (G.-Y.P.)
* Correspondence: chulki@pusan.ac.kr; Tel.: +82-051-510-2489

**Abstract:** Non-circular gears can maintain rotational motions of general gears and implement all varying rotational motions of the cam. They adjust the angular velocity of driven gear according to operating conditions and make precise changes in angular motion. The design of non-circular gears has not been sufficiently studied because of their particularity and complex design methods unlike spur gears. In the gutting section of the Squid Belly Opening and Gutting Machine (SBOGM), spur gears generate rotational impact due to constant angular velocities, causing noise and equipment damage; so, efficiency should be improved by varying sectional angular velocity. Therefore, we derived pitch curves by selecting angular velocity ratio considering operating environments, and the tooth profile was designed by calculating module for each section according to radius through theorical analysis for precise expression of angular velocity ratio. To confirm reliability of design, angular velocity ratio and structural safety of designed non-circular gears were verified using, commercial software, 'DAFUL 2020 R1'.

**Keywords:** non-circular gear; dynamic analysis; gear generation; tooth profile; DAFUL





## 1. Introduction

The Squid Belly Opening and Gutting Machine (SBOGM) is a machine that automates processes such as belly opening, gutting, and separation of the body and legs of squids, which were previously performed manually by people, and all processes of the machine are carried out through gears and chain transmission as shown in Figure 1. Among the processes, viscera removal is done by fixing the viscera of squids with the inhaler and then plucking them out. Since the inhaler always moves at the same speed, there is the problem that the viscera of squids may not be removed completely depending on the type and size of the squid, and thus only squids of a specific type and size can be processed. In addition, in the section where the inhaler rotates and returns to its original position for repetitive processes, the heavy weight of the inhaler causes impact during rotation, generating noise, and at a high velocity, there is a possibility of damage to the machine; so, it is not possible to increase the processing speed due to concerns about damage to the machine.

As a result, the quantity of squids processed per hour of the SBOGM is limited, resulting in reduced work efficiency and economy.

In order to solve the problem, it is necessary to control the speed of the gutting section, which was previously controlled by the spur gear and chain drive, and although we initially attempted to use an electric motor for speed control, we intend to use a non-circular gear instead of an electric motor because the use environment is a salty area near the sea and it is difficult to use an electric motor that has relatively higher failure rates than a mechanical motor. Jian [1] obtained the angular velocity graph of driven gear through the design of high-order elliptic gears and dynamics simulation using software MAPLE. Seong-Cheol Lee [2] applied the average module for an arbitrary number of teeth to all teeth and connected the tooth roots in a circular arc form in the design of non-circular gears according to the angular velocity ratio diagram. Zhou Kaihong [3] showed the theoretical

curve analysis of the angular velocity ratio and the consistency of analysis results for non-circular gears with an oval pitch curve by using the analysis software ADAMS. Marius VASIE [4] considered the simulation of rolling in a 2D space and verified the geometric and kinematic calculations performed in the design stage. Sang-Hoon Choi [5] studied optimal ranges of pressure angle, ratio of tooth number and radius of curvature of an elliptical gear. Dong-Woo Kang [6] represented the pitch curve of an elliptical gear in the form of a biarc and suggested a method of creating a tooth profile with an involute curve on each arc of biarc curves. Hao Xu [7] studied a general form of gears known as non-circular gears that can transfer periodic motion with variable speed through their irregular shapes and eccentric rotation centers. Sang-Hoon Choi [8] suggested one of references in applying non-circular gear to industrial plant, and suitable range of application by pressure angle curvature and angle ratio. Mircea Niculescu [9] introduced a new kinematics of the billet reheating furnace unloading door, by using a noncircular gear train in the drive mechanism. Marius Vasie [10] showed about the meshing conditions of non-circular gears. Fangyan Zheng [11] showed synthesizes still another special kind of indexing mechanism, using planetary non-circular gear train covering three gear trains with different gear types. Zhang Jian [12] proposed a simple algorithm for designing pitch curves for non-circular gears based on variable gear ratio functions. A A Lyashkov [13] proposed a geometrical model of solution to the task of profiling a non-circular gear, the centroid of which consists of interconnected arcs. D Mundo [14] presented a concept of epicyclical gear train able to generate a variable gear ratio law. CRISTESCU Ana [15] designed of multispeed gears pitch curves was taken into account, as the first important step in the gear design procedure. Specific original algorithms were dedicated to multispeed gear pitch curve modeling, based on the definition of gear transmission ratio variation law. Ming-Feng Tsay [16] derived a mathematical model of the tooth profile of a non-circular gear based on the approximate pitch curve. SABAH KHAN [17] measured the transmission error to determine the noise and vibration of non-circular using 'ANSYS'. A Cristescu [18] investigated multi-speed gear generation procedures and gear bending conditions. However, previous studies used highly similar pitch curve design methods and dealt with the cases where the pitch curve is elliptical or relatively gentle. Hence, there has been a lack of research on module selection and tooth profile design considering the operating environment as well as the curvature of the pitch curve of non-circular gears in relation to the characteristics of non-circular gears with angular velocity ratio varying continuously from moment to moment unlike general spur gears.

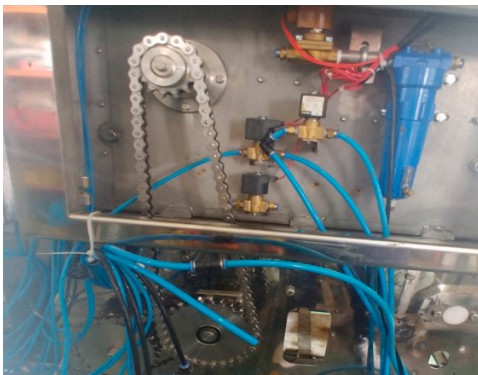

**Figure 1.** Conventional driving method of the SBOGM.

Therefore, in this study, we intend to design a non-circular gear that can be applied to a special use environment where the velocity should be decreased to reduce impact in the rotation section while the transportation speed of the inhaler should be rapidly increased to increase the force for viscera removal. The ratio of angular velocity was selected in consideration of the operating environment and expressed as a function for each section. The curvature of each section of the derived pitch curve was calculated, and modules

were calculated for each section, based on the transfer load and radius of the pitch. The driving gear was designed by applying the involute tooth shape determined for each section, and the driven gear for the designed driving gear was generated using the driven gear generation program. Finally, the ratio of angular velocity and structural safety of the designed non-circular gear were verified by conducting a flexible body dynamics analysis using 'DAFUL' according to design procedures shown in Figure 2.

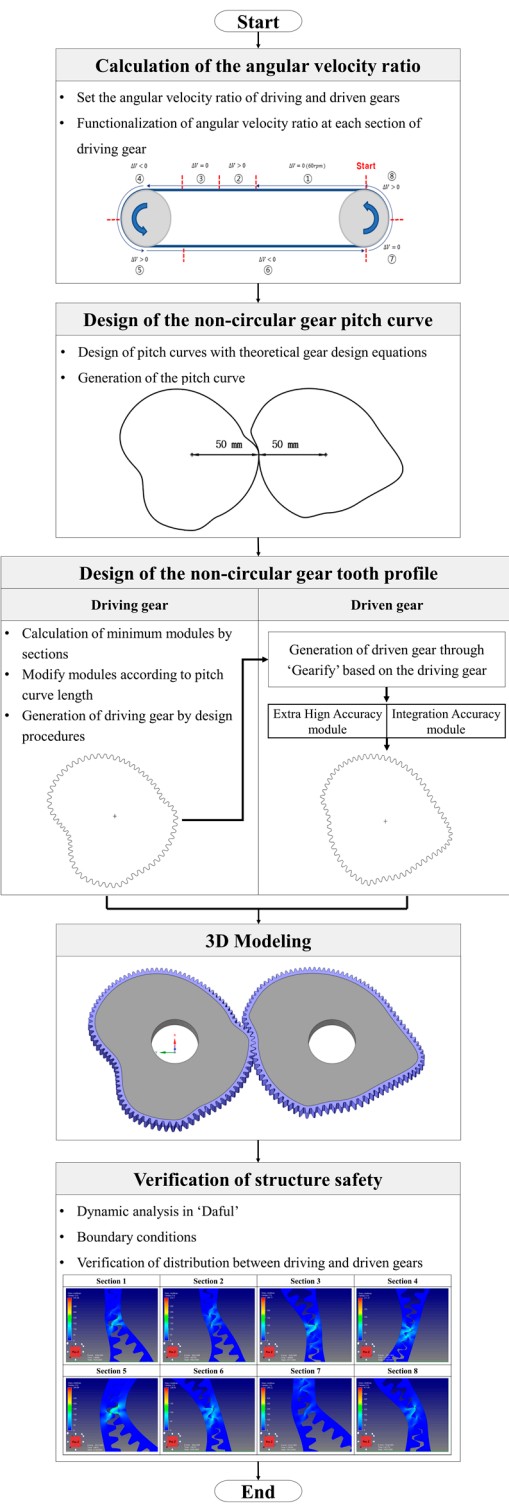

**Figure 2.** Configuration of non-circular gear design procedure.

## 2. Design of the Non-Circular Gear

### 2.1. Design of the Pitch Curve

#### 2.1.1. Design of the Angular Velocity Ratio

In order to design the pitch curve, the angular velocity ratio of the viscera removal section needs to be selected. In the existing equipment, the section was operated through a spur gear at a constant angular velocity of 60 rpm, but to reduce rotational impact and improve work efficiency, the velocity was increased by 30% compared to that of the existing equipment in the section where the actual gutting process occurs (②~③), while the velocity was decreased by 30% compared to that of the existing equipment to reduce rotational impact in the section in which the inhaler rotates (④~⑤, ⑦~⑧). The entire acceleration and deceleration sections are as shown in Figure 3.

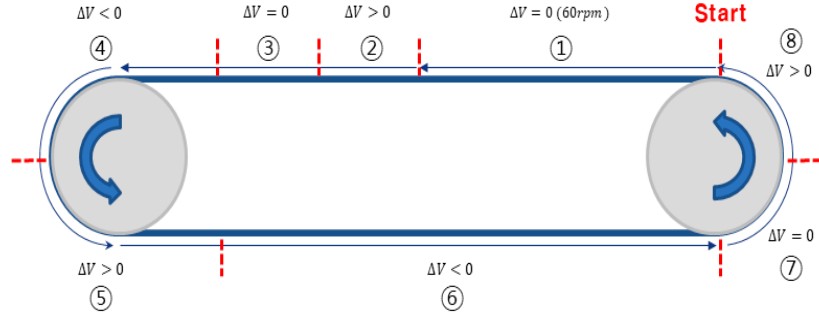

**Figure 3.** Changes in sectional velocity in the gutting section of the SBOGM.

The graph of the angular velocity ratio of the driven gear to the driving gear according to the selected acceleration and deceleration sections is shown in Figure 4. In the case of non-circular gears, the ratio of the driving gear to the driven gear must be designed with a natural number, and the rotation starting point of the driving gear and driven gear must always be constant. To satisfy these conditions, the areas of deceleration and acceleration sections in the angular velocity ratio graph of the driving gear must be identical.

$$\sum_{1}^{k} \int_{n-1}^{n} f'(\theta) = 1, 2, 3, \cdots (k : \text{number of spans}) \tag{1}$$

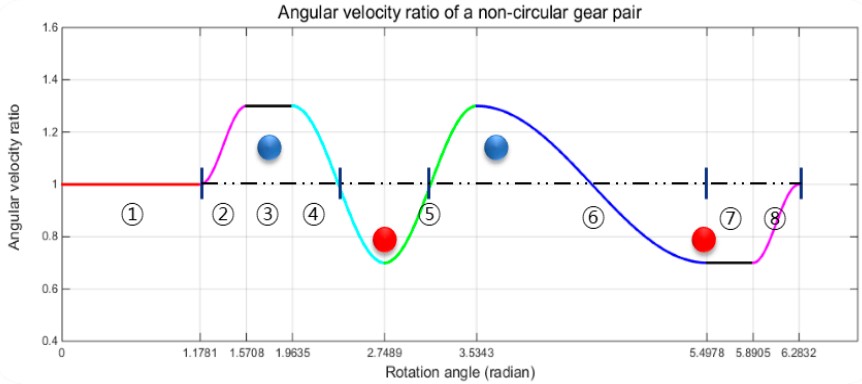

**Figure 4.** Sectional angular velocity ratio of driving gear to driven gear.

Here, to express each section as a single function, a combination of functions for each section was selected as shown in Table 1 so that the average angular velocity ratio per period for each section would be 1. The average angular velocity ratio was set as 1, and 'h' was set as 0.3 to increase or decrease the angular velocity by 30% in the acceleration/deceleration section.

**Table 1.** Boundaries of sectional angular and each function.

| Section | $\theta$ [Radian] | $f'(\theta)$ |
|---|---|---|
| ① | $\left[0, \frac{3}{16}2\pi\right]$ | $f_1' = a$ |
| ② | $\left[\frac{3}{16}2\pi, \frac{4}{16}2\pi\right]$ | $f_2' = a + \frac{h}{2}\cdot(1 + \cos(8x))$ |
| ③ | $\left[\frac{4}{16}2\pi, \frac{5}{16}2\pi\right]$ | $f_3' = a + h$ |
| ④ | $\left[\frac{5}{16}2\pi, \frac{7}{16}2\pi\right]$ | $f_4' = $ $a + h - h\cdot\left(1 + \cos\left(4x + \frac{\pi}{2}\right)\right)$ |
| ⑤ | $\left[\frac{7}{16}2\pi, \frac{9}{16}2\pi\right]$ | $f_5' = $ $a - h + h\cdot\left(1 + \cos\left(4x - \frac{\pi}{2}\right)\right)$ |
| ⑥ | $\left[\frac{9}{16}2\pi, \frac{14}{16}2\pi\right]$ | $f_6' = a + h - $ $h\cdot\left(1 + \cos\left(\frac{8}{5}x - \frac{4}{5}2\pi\right)\right)$ |
| ⑦ | $\left[\frac{14}{16}2\pi, \frac{15}{16}2\pi\right]$ | $f_7' = a - h$ |
| ⑧ | $\left[\frac{15}{16}2\pi, 2\pi\right]$ | $f_8' = a - h + \frac{h}{2}\cdot(1 + \cos(8x))$ |

### 2.1.2. Design of the Pitch Curve

In non-circular gears, the rotation angle ($\varphi$) of the driven gear can be obtained differently from the rotation angle ($\theta$) of the driving gear with a constant angular velocity, and the rotation angle ($\varphi$) of the driven gear is expressed by Equation (2).

$$\varnothing = f(\theta) \tag{2}$$

Here, $\frac{d\varnothing}{d\theta} = f'(\theta)$ represents the angular velocity ratio for the rotation angle ($\varnothing$) of the driven gear to the rotation angle ($\theta$) of the driving gear.

As shown in Figure 5, if the distance ($c$) between the rotation axes of the driving and driven gears is constant as shown in Equation (3) and the gear ratio is 1:1, since the pitch curves are in rolling contact with each other, rolling distance $l_1$ of the pitch curve of the driving gear is equal to the rolling distance $l_2$ of the pitch curve of the driven gear. Therefore, an infinitesimal change in each rolling distance at an arbitrary pitch point can be expressed by Equation (4).

$$r_1(\theta) + r_2(\theta) = c \tag{3}$$

where,

$$r_1 d\theta = r_2 d\varnothing \tag{4}$$

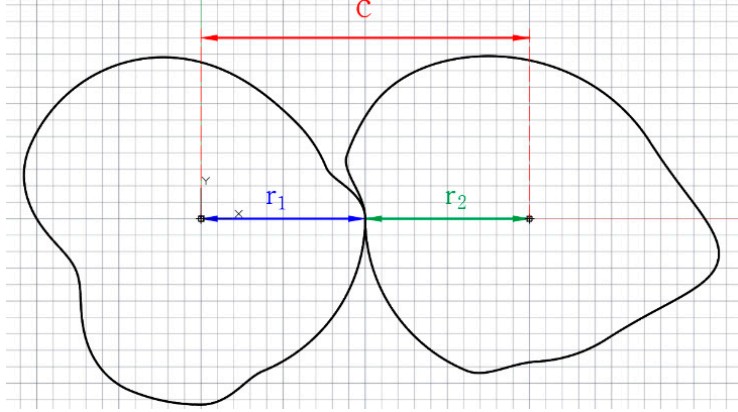

**Figure 5.** Pitch curves of driving gear and driven gear.

By combining Equations (3) and (4), $r_1$ and $r_2$ are derived as follows:

$$r_1(\theta) = c \cdot \frac{\frac{d\varnothing}{d\theta}}{1 + \frac{d\varnothing}{d\theta}} = c \cdot \frac{f'(\theta)}{1 + f'(\theta)} \quad r_2(\theta) = c \cdot \frac{1}{1 + \frac{d\varnothing}{d\theta}} = c \cdot \frac{1}{1 + f'(\theta)} \tag{5}$$

The length of the pitch curve of each section is given by Equation (6).

$$S_i = \int_{\theta_{i-1}}^{\theta_i} \sqrt{(r_1')^2 + (r_2')^2} \, d\theta \tag{6}$$

The total length of the pitch curve is calculated by Equation (7).

$$S = \sum_{i=1}^{i} S_i \tag{7}$$

In other words, if the angular velocity ratio $f'(\theta)$ and the inter-axis distance 'c' are determined, the pitch curve can be determined from the derived $r_1(\theta)$ and $r_2(\theta)$, and its results were shown in Table 2.

**Table 2.** Specifications of driving gear and driven gear according to the sections.

| Section | 1 | 2 | 3 | 4 | 5 | 6 | 7 | 8 |
|---|---|---|---|---|---|---|---|---|
| Pinion rotation angle (°) | 0~67.5 | 67.5~90 | 90~112.5 | 112.5~157.5 | 157.5~202.5 | 202.5~315 | 315~337.5 | 337.5~360 |
| Gear rotation angle (°) | 0~67.5 | 67.5~93.37 | 93.37~115.87 | 115.87~160.89 | 160.89~205.87 | 205.87~318.04 | 318.04~340.54 | 340.54~360 |
| Radius of Driving Gear (mm) | 50 | ↑ | 66.67 | ↓ | ↑ | ↓ | 33.33 | ↑ |
| Radius of Driven Gear (mm) | 50 | ↓ | 33.33 | ↑ | ↓ | ↑ | 66.67 | ↓ |
| Pitch curve length (mm) | 58.90 | 22.17 | 22.20 | 28.60 | 27.18 | 29.09 | 114.75 | 20.42 |
| Total pitch curve length (mm) | | | | 323.31 | | | | |

### 2.1.3. Derivation of the Pitch Curve

If $f'(\theta)$ obtained in Table 1 is substituted into Equation (5), the coordinates of the pitch curves of the driving and driven gears can be obtained. In this study, the inter-axis distance was set as 100 mm according to the actual operating environment, and for data reduction for the ease of 2D and 3D modeling, the error of the pitch curve according to the number of coordinate points was examined. When using 3600 coordinates obtained in every case of $\theta = 0.1°$, the maximum distance error was generated as $1 \times 10^{-6}$ mm or less.

### 2.2. Tooth Profile Design

Unlike general spur gears that maintain the same velocity, non-circular gears have continuously changing angular velocity of the driven gear during one rotation of the driving gear, so it is difficult to design a non-circular gear with the same module. So, minimum module was determined through theoretical equation for each section, satisfied with structural safety and obtained precision of angular velocity described.

### 2.2.1. Calculation of Minimum Module

In order to determine module for each section, it is necessary to calculate the minimum module that can be used in the operating environment. In the case of non-circular gears, the distance from the center of rotation to the pitch point continuously changes during the rotation of gears which are engaged with each other. The largest changes in angular velocity occur in the section with the largest curvature, and in order to accurately represent the angular velocity, module was based on the position with the smallest radius in the

driving gear, and a sufficient safety factor was given in consideration of the specificity of non-circular gears. For the transfer load ($W$), the minimum module is given by Equation (8).

$$m = \frac{w}{s f_v b y \sigma_b} \tag{8}$$

The minimum module is obtained from Equation (8), where $w$ is the transfer load, $s$ is the safety factor, $f_v$ is the velocity factor, $y$ is the tooth form factor, and $\sigma_b$ is the bending stress in gear tooth.

The torque load required in the field is 50′000 N·mm in the viscera removal section, and the transfer load $W_n$ is calculated by Equation (9). Each interval of driving gear is shown in Figure 6, and theoretically calculated module is shown in Table 3.

$$W_n = \frac{w}{\cos(\alpha)} \div R_{\min} \tag{9}$$

$$m = \frac{p}{z\pi} (z = 1, 2, 3, \dots \text{Natural number}) \tag{10}$$

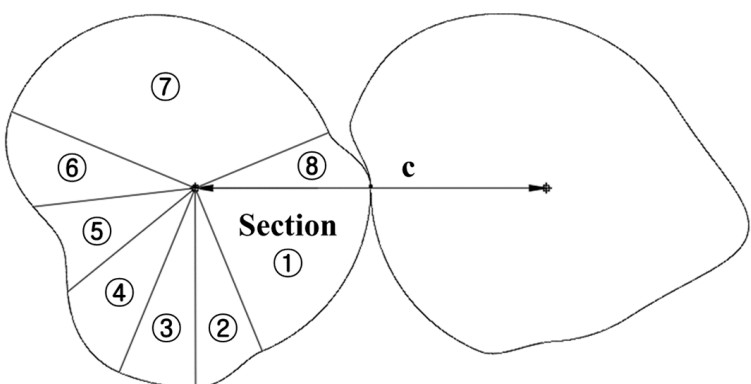

**Figure 6.** Pitch curves divided by sections (c = 100 mm).

**Table 3.** Tooth profile design according to each section.

| Angle (°) | Length (mm) | Theory Min Module | Number of Teeth | Module |
|---|---|---|---|---|
| 0°~67.5° | 58.90 | 1.2 | 15 | 1.25 |
| 67.5°~90° | 22.17 | 1.2 | 5 | 1.41 |
| 90°~112.5° | 22.20 | 1.316 | 5 | 1.41 |
| 112.5°~141° | 28.60 | 1.08 | 8 | 1.14 |
| 141°~173.5° | 27.18 | 0.96 | 9 | 0.96 |
| 173.5°~202.5° | 29.09 | 1.08 | 8 | 1.15 |
| 202.5°~337.5° | 114.75 | 0.96 | 38 | 0.96 |
| 337.5°~360° | 20.42 | 0.96 | 6 | 1.08 |

The minimum module obtained from Equation (8) was adjusted by Equation (10) because the number of teeth within each section must be a natural number.

Design procedures of non-circular gear are composed of 4 steps and are as follows.

1. Creation of involute tooth according to the modules calculated (Figure 7a).
2. Equal division of pitch curve at each section according to the modules adjusted (Figure 7b).
3. Arrangement of centers (pitch points) of circle in the equal points by translation and rotation (Figure 7c).
4. Connection of roots between designed teeth (Figure 7d).

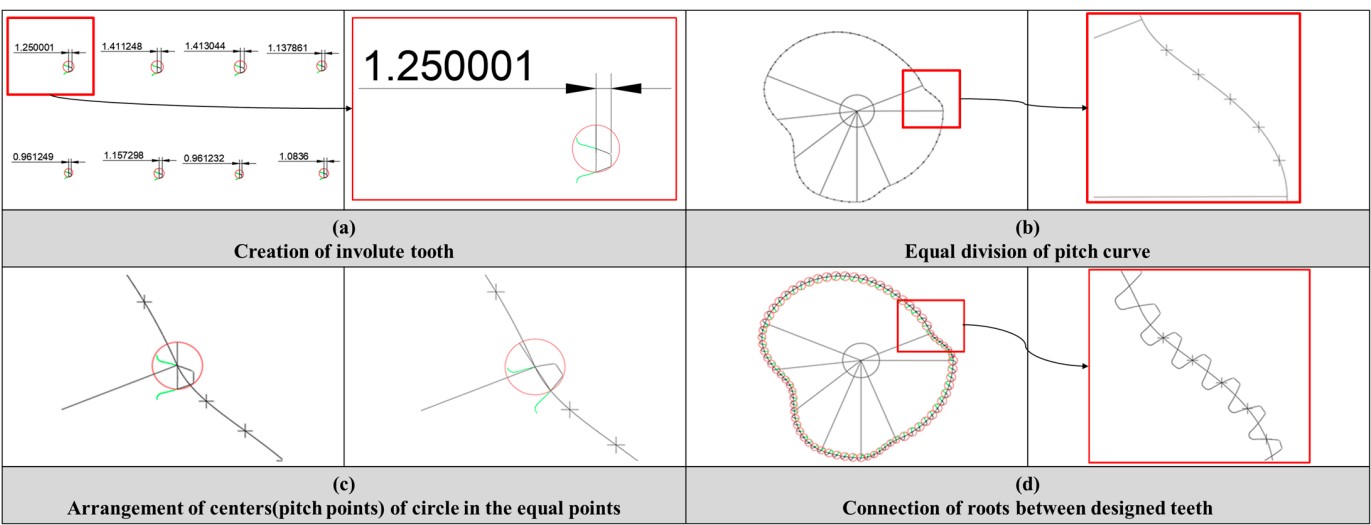

**Figure 7.** Design procedures of non-circular gear.

Proceeding in the above order, the final driving gear design is completed. The shape of the final designed driving gear is shown in Figure 8.

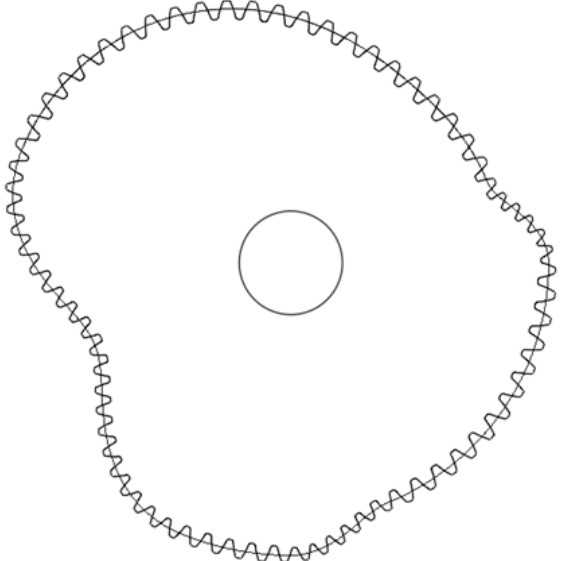

**Figure 8.** The shape of the final designed driving gear.

### 2.2.2. Design of the Tooth Profile of the Driven Gear

The design of the driven gear was carried out using the gear design program 'Gearify' to ensure that the driven gear is precisely engaged with the driving gear while it is rotating. The pitch curve of the driving gear and its final shape were imported into 'Gearify' as shown in Figure 9.

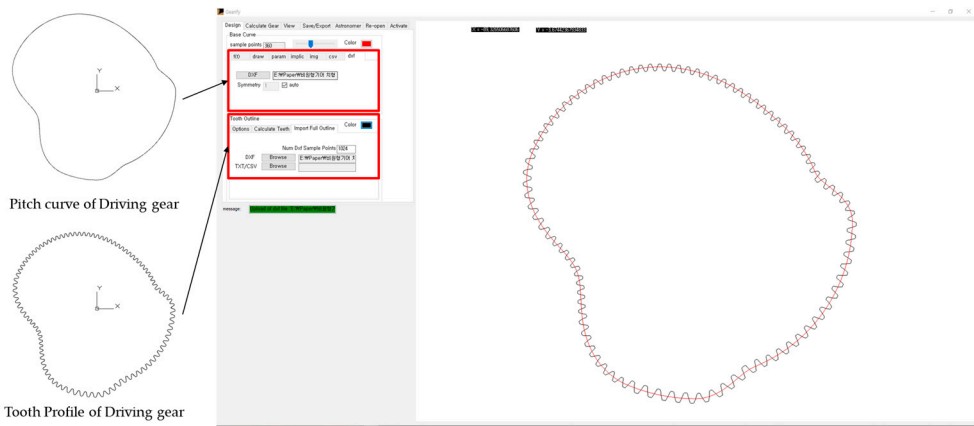

Pitch curve of Driving gear

Tooth Profile of Driving gear

**Figure 9.** Insertion of design information for driving gear.

'Extra High Accuracy' was selected to derive a precise design of the driven gear, and its teeth were smoothly connected by setting 'Integration Accuracy' as high as shown in Figure 10.

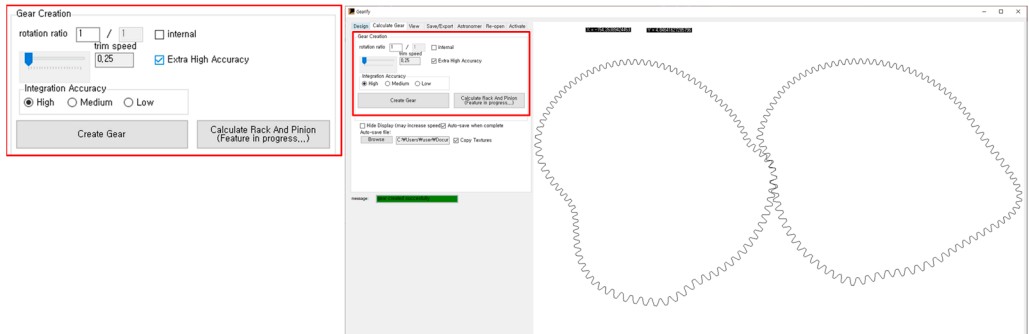

**Figure 10.** Setting of driven gear for precise design.

If the designed pitch curve and gear curve of the driving gear are input into the program which is proceeded with, the driven gear is generated through 'Gearify' and is shown in Figure 11.

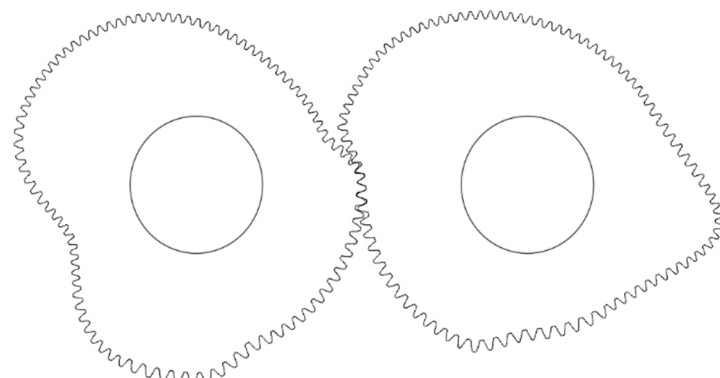

**Figure 11.** Final driven gear generated according to the designed driving gear.

### 3. Verification of Structural Safety of Non-Circular Gears

*3.1. 3D Modeling*

Flexible body dynamic analysis was conducted to examine the angular velocities of designed driving and driven gears and their structural safety was confirmed using the dynamic analysis program 'DAFUL'; analytical modeling was carried out using 'SpaceClaim'.

Tooth thickness (10 mm) was the same as the chain gear and the spur gear used in the actual field, and gears were divided into two parts (tooth part and body part) in order to reduce analysis time, as shown in Figure 12, offsetting pitch curve.

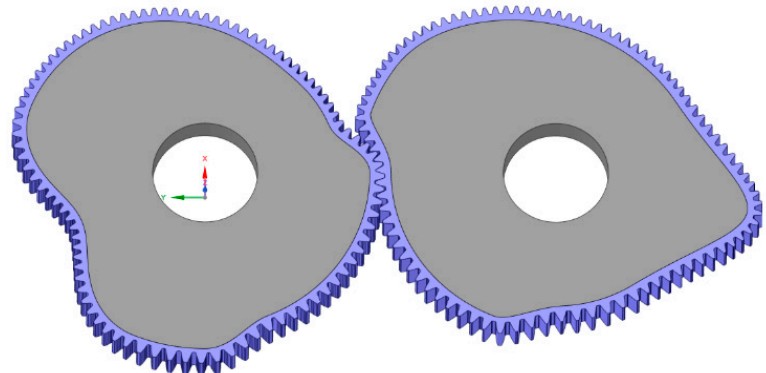

**Figure 12.** Three-dimensional modeling of non-circular gear.

*3.2. Boundary Conditions*

Based on analytical modeling, the mesh for the tooth part was created with 'Ansys-Workbench'. The mesh was controlled by edge sizing and the sweep method, and a mesh consisting of 231′612 elements and 1′226′260 nodes were generated through the analysis of mesh quality as shown in Figure 13. The generated mesh data was exported as an '.inp file' to transfer it to 'Daful'.

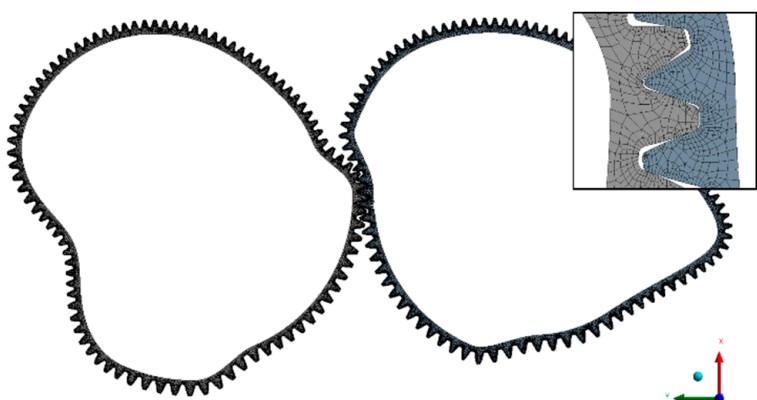

**Figure 13.** Generation mesh data of non-circular gear.

After importing the generated 'Mesh file' from 'Daful' to conduct dynamic analysis, the 'Joint condition' was given so that the driving and driven gears could rotate around the rotation axis. Also, considering the actual operating environment, the rotational speed of 60 rpm was applied for the driving gear and the torque load of 50′000 Nmm was applied for the driven gear. For the convergence of analysis, as shown in Figure 14, rotational velocity and torque load were set so that they would gradually increase with time by using the 'Step function', as shown in Figure 15.

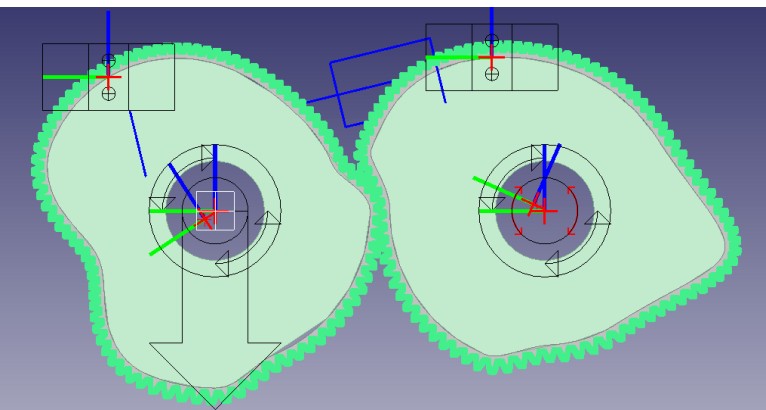

**Figure 14.** Boundary condition setting for dynamic analysis.

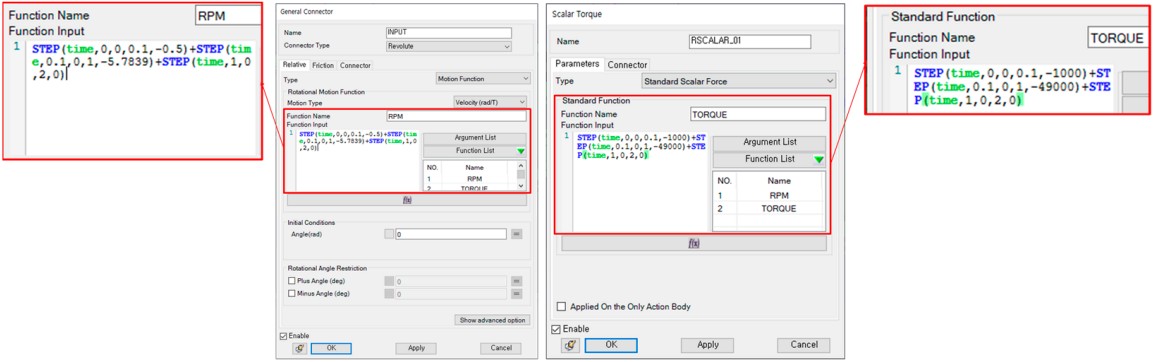

**Figure 15.** Setting the rotational velocity of the driving gear and the rotational load of the driven gear.

In the case of contact conditions, as shown in Figure 16, the 'Friction Coefficient' was set as 0.08 according to the general friction conditions of gears. The total analysis time was 3 s, and the 'Output Step' was set as 2000 to obtain detailed analysis results.

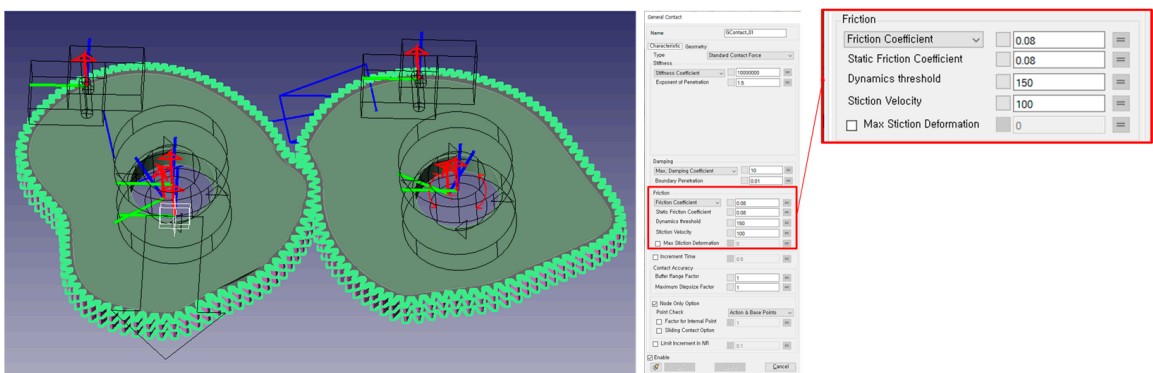

**Figure 16.** Contact condition setting for non-circular gears.

### 3.3. Analysis Results

The angular velocity ratio during one revolution of the driving gear was extracted through 'DAFUL' analysis, based on the time points when the velocity was sufficiently developed.

The results are shown in Figure 17, and it can be seen that analysis results of angular velocity ratio are identical to the graph of angular velocity ratio obtained by theoretical calculations.

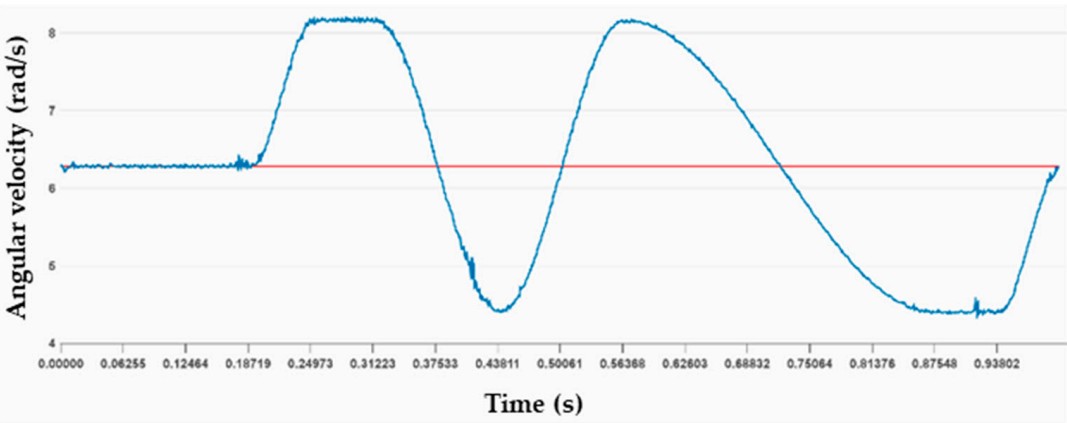

**Figure 17.** The angular velocity result through dynamic analysis.

In order to verify the structural safety, Von-Mises stresses of driving and driven gears were extracted. Figure 18 show the stress distribution of each section during one revolution of the driving gear and driven gear, respectively. In the sections of corresponding to circular motion with a constant speed, Von-Mises stress is relatively constant, but in the sections where the angular velocity of the driven gear increases or decreases, stress is higher than in the constant velocity section.

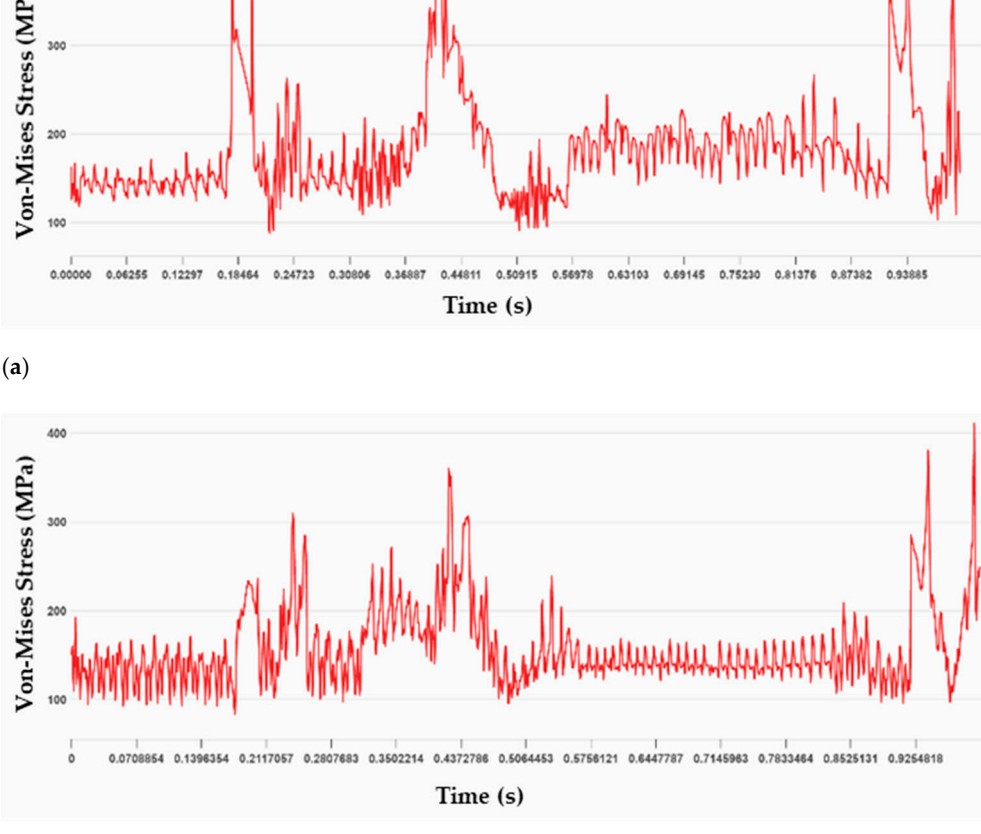

(**a**)

(**b**)

**Figure 18.** Stress distributions during one revolution of the driving and driven gears. (**a**) Driving gear and (**b**) Driven gear.

Table 4 showed the maximum equivalent stress at each section of the driving and driven gears considering safety factor 1.2, and the maximum stress in Section (8) was 411.45 MPa, which was lower than the yield strength (SUS440C) as shown in Figure 19, i.e., 445 MPa so that structural evaluation of non-circular gear was considered to be safe.

**Table 4.** Max. equivalent stress result for each section.

| | Range (°) | Max. Stress (MPa) | |
| --- | --- | --- | --- |
| | | **Driving Gear** | **Driven Gear** |
| Section 1 | 0~67.5 | 375.36 | 201.99 |
| Section 2 | 67.5~90 | 372.70 | 309.43 |
| Section 3 | 90~112.5 | 256.76 | 284.77 |
| Section 4 | 112.5~141 | 218.25 | 271.27 |
| Section 5 | 141~173.5 | 399.89 | 360.14 |
| Section 6 | 173.5~202.5 | 193.58 | 238.99 |
| Section 7 | 202.5~337.5 | 399.22 | 285.49 |
| Section 8 | 337.5~360 | 387.65 | 411.45 |

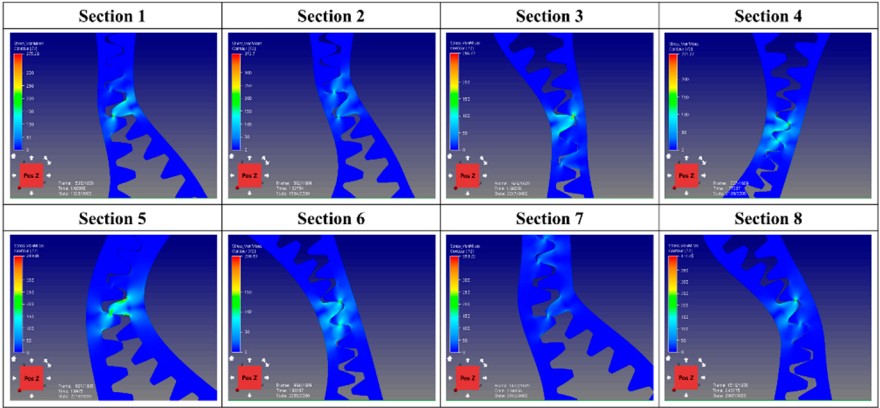

(**a**)

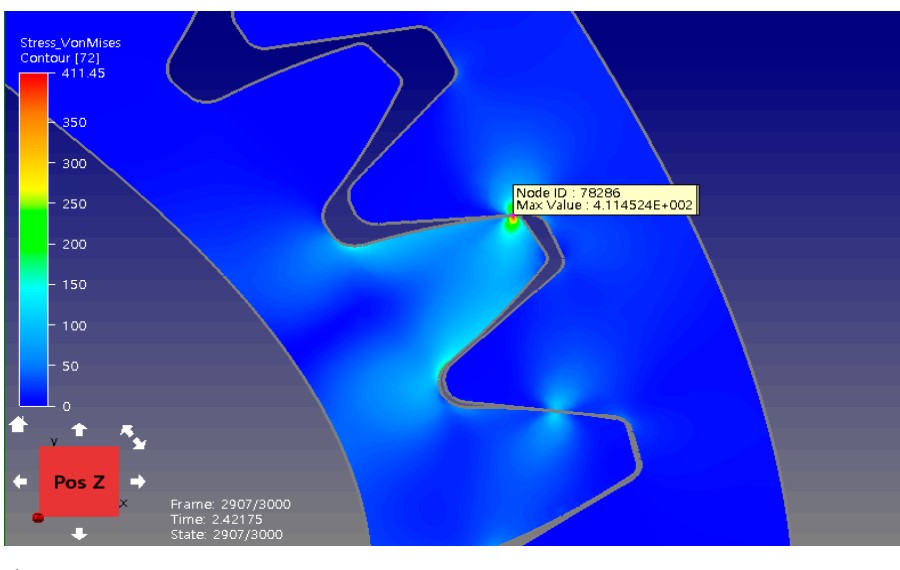

(**b**)

**Figure 19.** The simulation results of structural evaluation of non-circular gears. (**a**) The maximum Von-Mises stress in each section. (**b**) The maximum Von-Mises stress at the Section (8).

## 4. Conclusions

In this study, we intend to design non-circular gears that can be applied to a special environment where the velocity should be decreased to reduce impact in the rotation section while the transportation speed of the inhaler should be rapidly increased to increase the force for viscera removal. And the ratio of angular velocity and structural safety of the designed non-circular gear were verified by conducting a flexible body dynamics analysis using 'DAFUL'. The results of the study are summarized as follows.

(1) To improve the efficiency of the processes of the existing Squid Belly Opening and Gutting Machine (SBOGM) and to reduce impact during rotation, non-circular gears alternative to the chain gear and spur gear were developed for the SBOGM.

(2) Driving gear was designed according to the tooth design procedures suggested for non-circular gears, and the driven gear was generated through its program 'Gearify' based on the designed driving gear.

(3) The angular velocity ratio during one revolution of the driving gear was extracted through 'DAFUL' analysis, whose results are identical to the graph of angular velocity ratio obtained by theoretical calculations.

(4) Flexible body dynamics analysis was conducted using the commercial program 'DAFUL', and the maximum Von-Mises stress at the Section (8) was 411.45 MPa, which was lower than the yield strength of 445 MPa, so the structural safety was considered to be safe.

**Author Contributions:** Conceptualization, H.-S.J. and C.-H.L.; formal analysis, H.-S.J.; investigation, C.-H.L.; methodology, G.-Y.P. and C.-H.L.; writing—review and editing, H.-S.J and C.K. All authors have read and agreed to the published version of the manuscript.

**Funding:** This work was supported by a 2-Year Research Grant of Pusan National University.

**Institutional Review Board Statement:** Not applicable.

**Informed Consent Statement:** Not applicable.

**Data Availability Statement:** Not applicable.

**Conflicts of Interest:** The authors declare no conflict of interest.

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
