# Peer review of "Study on Design of Non-Circular Gears for Speed Control of the Squid Belly Opening and Gutting Machine (SBOGM)"

_applsci, doi:10.3390/app11073268_

Round 1

Reviewer 1 Report

A high-quality article with great application values.

Two Notes:

- no description of the coordinate axes in fig. 17 and 18

- the bibliography contains too few references.

Author Response

Thank you for your peer review.

Reviewer 2 Report

The central idea of the work - Study on design of non-circular gears for speed control of the squid belly opening and gutting machine (SBOGM).

The authors of the study noted the lack of sufficient design research related to non-circular gears used to eviscerate SBOGM. In the currently used spur gears, they derived the pitch curves, selecting the angular velocity ratio taking into account the working environment, and designed the tooth profile for each section according to the radius using a calculation module. In order to verify the theoretical assumptions, the angular velocity coefficient and structural safety of the designed non-circular gears were checked using the commercial software "DAFUL 2020 R1".

The obtained data is very interesting and brings new knowledge in the field of non-circular gears.

In my oppinion scientific importance of the paper is very high.

The structure of the article is very transparent. Proposed methodology and conclusion are good discribe.

Note to the authors of the work:
1. you should put a space between the word and the bracket or unit in the lines: 46, 50, 109, 138, 158, 159, 193, 194, 205, 211, 227, 228, 247 and 248.
2. in row 199 two commas in a single digit.

Author Response

Thank you for your peer review. 
